# Changes in Risk in Medium Business Plating and Paint Manufacturing Plants following the Revision of the Korean Chemical Accident Prevention System

**DOI:** 10.3390/ijerph182211982

**Published:** 2021-11-15

**Authors:** Hyo Eun Lee, Min-Gyu Kim, Seok J. Yoon, Da-An Huh, Kyong-Whan Moon

**Affiliations:** 1Department of Health Science, Korea University, Anam-ro 145, Seongbuk-gu, Seoul 02841, Korea; chokbab@naver.com; 2Department of Health and Safety Convergence Science, Korea University, Anam-ro 145, Seongbuk-gu, Seoul 02841, Korea; rbalsrla1@naver.com; 3Institute of Health Science, Korea University, Anam-ro 145, Seongbuk-gu, Seoul 02841, Korea; ehslab@naver.com (S.J.Y.); black1388@korea.ac.kr (D.-A.H.); 4Department of Health and Environmental Science, Korea University, Anam-ro 145, Seongbuk-gu, Seoul 02841, Korea; 5BK21 FOUR R&E Center for Learning Health System, Korea University, Anam-ro 145, Seongbuk-gu, Seoul 02841, Korea

**Keywords:** Preparation of Off-Site Consequence Analyses, Chemicals Control Act, risk assessment, paint manufacturing plant, plating industry plant

## Abstract

Chemical accidents can occur anywhere. The need for chemical management in Korea was realized following the 2012 Gumi hydrofluoric acid accident in 2012. The Chemicals Control Act was enacted in 2015. This system evaluates the risks (high, medium, low) and consequent safety management at all plants that handle hazardous chemical substances. However, the system was criticized as excessive when most plants were designated high-risk without considering their size. Thus, laboratories and hospitals handling very small quantities were subject to regulation. Accordingly, in 2021 Korea revised the system to include off-site consequence analyses and a Korean-style risk analysis. Plants handling very small quantities, such as laboratories and hospitals, were exempt from regulation. In this study, changes in risk were examined for four medium-sized plating and paint manufacturing plants. Under the previous system, all four factories were judged as high-risk groups. In particular, the paint manufacturing plant A, which has an underground storage tank, received a medium risk like the plating plant C, although the possibility of a chemical accident was lower than that of other plants. However, in the changed system, all plants were changed to the low-risk group. In the Korean-style risk analysis, it is possible to see at a glance what is lacking in the plants, such as cooperation between local residents and local governments and the construction of safety facilities according to the type of accident scenario. The revised system is a reasonable regulation for medium business plants.

## 1. Introduction

### 1.1. History of Chemical Accidents

Chemical accidents are an inevitable consequence in industries that use chemicals and inevitably occur in the development of technology in the chemical industry. Accordingly, many business owners and engineers have conducted numerous studies to prevent chemical accidents and minimize damage in the case of chemical accidents [1].

Various methods to prevent accidents include safety-enhancing technological and organizational changes, compilation of chemical accident databases, barriers, and adequate incentives for the prevention of chemical accidents, and economic incentives to improve chemical safety. However, the method adopted by most countries that is nationally applicable is to strengthen the system and regulations for chemical safety management [2].

Concerning the history of chemical accident prevention systems, the Seveso Directive was enacted following the 1976 dioxin leak in Seveso, Italy [3]. Subsequently, the 1984 methyl isocyanate (MIC) leak in Bhopal, India, highlighted that a chemical accident was not necessarily confined within the plant, but rather could have a significant and devastating impact on local residents and the surrounding environment. The Bhopal accident killed 2500 to 6000 people died, with another approximately 500,000 residents afflicted with chronic obstructive pulmonary disease. The Bhopal disaster prompted the establishment of standards for process safety management (PSM) by international organizations. PSM systems have been legislated in many countries worldwide [4].

### 1.2. Regulations for the Prevention of Chemical Accidents in the U.S. and Korea

In the United States, PSM is managed by the Occupational Safety and Health Administration. Its main contents include on-site and off-site safety regulations. These include using written operating procedures, providing employee training, ensuring ongoing mechanical integrity of equipment, and analyzing and controlling process hazards.

PSM has been criticized for focusing on workplace safety and worker protection. Accordingly, the U.S. Environmental Protection Agency (EPA) added off-site consequence analysis (OCA) pertaining to the simulation of chemical accidents beyond PSM, and the risk management plan (RMP) system for the analysis and response to risks of accidents [5].

The PSM system was introduced in Korea in accordance with the Occupational Safety and Health Act of 1996. The legislation aimed at prevention of worksite chemical accidents. Prior to this, chemical accident prevention focused on the safety of workers in the workplace. This was because chemical-related accidents in Korea have been mainly related to occupational events, such as the Wonjin Rayon incident in the 1990s.

The hydrofluoric acid accident in Gumi in 2015 highlighted that chemical accidents in Korea not only affect the environment and people inside plants but also pollute the surrounding environment and detrimentally affect local residents. Accordingly, the Korea Ministry of Environment introduced the U.S. RMP, OCA, and RMP systems, and enacted the Chemicals Control Act.

The legislation goes beyond the internal regulations and safety management aspects of the PSM. By simulating chemical accidents, it can be virtually determined which safety equipment should be installed to minimize the accident risk and damage. The chemical accident scenarios are used to perform a risk ranking to determine which chemical facility is the most dangerous. In addition, in connection with local business sites, a community network is formed for the evacuation of local residents and environmental protection in the event of a chemical accident, and joint training is planned.

In Korea, the Chemical Control Act was introduced from 2015 to 2021. Business sites nationally are managed by classifying them into high-, medium-, and low-risk plants based on the scenarios and amounts of chemicals used. However, even hospitals and laboratories that use only 1 g of chemicals are included in the regulation. This has been criticized as excessive administration and regulation. The possibility of a simple chemical accident is easily determined through a scenario. The estimation of a chemical accident by the scenario modeling tool is problematic, since the influence of chemical compounds can vary markedly.

For example, there are many accident scenarios for hydrochloric acid, which has many uses. Thus, a small plant that handles only hydrochloric acid may pose a greater risk than other (even larger) plants that handle flammable substances such as toluene and benzene, which are likely to have a smaller scenario range than hydrochloric acid. This example suggests that it is impossible to adequately respond and appropriately to the size of the workplace, because medium-size business plants can be in the high-risk group along with large-size plants [6].

In particular, while RMP and PSM are regulated in the U.S. mainly by large companies, in Korea, regulations with considerable legal power have been introduced, even for small- and medium-sized plants. To address the drawbacks, in 2021 Korea abolished the OCA system and introduced the Preparation of Off-Site Consequence Analyses (POCA) system. POCA includes aspects of the OCA and RMP systems, forming a single comprehensive regulation that subdivides the risk determination grades between large companies and medium companies [7].

This study compares the overall risk of plants in the chemical accident prevention system, the 2015 version of the OCA, and the newly implemented POCA system. The feasibility of the POCA system for small- and medium-sized plants was evaluated and limitations were identified.

## 2. Materials and Methods

### 2.1. Changes in the Chemical Accident Prevention System

According to the Chemicals Control Act of 2015, plants in all industries that handle chemical substances must submit OCA or RMP data depending on the amount of chemicals used.

In Korea, 97 of approximately 1000 hazardous chemical substances are designated as “chemicals requiring preparation for accidents”. The chemicals are considered highly capable of causing chemical accidents due to their potent acute toxicity or explosiveness, with the potential for considerable damage.

In the revised legislation, an OCA should be submitted by plants that handle all hazardous chemicals. In contrast, the RMP is submitted by plants that handle more than the specified amount of chemicals that require accident preparation plans [8].

As discussed above for laboratories and hospitals, the previous regulatory system in Korea was excessive, requiring filing of an OCA report even if only small amounts of hazardous chemicals, such as hydrochloric acid and nitric acid, are handled.

Secondly, in the former OCA legislation, many low- and medium-sized businesses were judged to be high-risk plants. This included 70% of the plants in Gyeonggi-do, an industrial complex where many small- and medium-sized businesses are concentrated in Korea. Facilities in the petroleum industry, a typical large-scale industry, were deemed to be equally high-risk as small- and medium-sized businesses, such as plating and paint manufacturing [9].

For such high-risk plants, safety inspections supervised by the Ministry of Environment must be performed once every four years. In the chemical accident prevention system, the excessive number of high-risk plants means excessive regulation, which may lead to administrative and personnel waste.

Thirdly, a summary of chemical accident information was provided to local residents and cooperation with local governments to prevent chemical accidents and provide emergency evacuation drills. Since this was included only in the RMP, plants that only submitted OCA data were not appreciably different from PSM concerning internal risk analysis and self-management.

By introducing the chemical accident prevention plan system in 2021, laboratories that handle small amounts of hazardous chemicals and hospitals that only handle some special products (such as ethylene oxide gas and formalin) were excluded from the chemical accident prevention system. In addition, a Korean-style plant risk analysis (K-risk analysis) was introduced to improve the risk distribution of plants. In the revised legislation, the system for emergency evacuation drills and information provision to local residents was expanded to small- and medium-sized businesses, and the chemical accident prevention system was reorganized [10].

Figure 1 shows the changes in the chemical accident prevention system in Korea from 2021. Under the previous system, all plants handling hazardous chemicals were included in the chemical accident prevention system. However, in the changed system, it has been reasonably changed, such as excluding plants that handle small quantities, laboratories, and hospitals.

### 2.2. Selection of Target Research Plants

A key aspect of the changed chemical accident prevention system used in this study is risk analysis. Risk analysis in the former OCA report was divided into high-, medium-, and low-risk categories of plants according to the workplace accident scenario.

The level of risk depends on the characteristic hazards of the chemicals handled in the factory and the amount of chemicals stored. The risk level also depends on whether the factory has facilities that can minimize damage in the event of an accident, such as water curtains or fire extinguishing facilities, and whether or not they have facilities, such as gas detectors and leak detectors, to detect early-stage abnormalities in chemical facilities. This approach can determine the most dangerous location in plants and which chemical facility is the most dangerous among various chemical facilities given a worst-case scenario. However, a limitation of this method is that it does not reflect the characteristics of an entire plant with few or many handling facilities.

For example, a plant with a hydrogen chloride storage tank is classified as a high-risk plant. In contrast, a plant with 10 tanks for storing solvents, such as toluene or xylene, can be classified as low- or medium-risk as a result of modeling compared to a plant that has one tank of hydrogen chloride that can be considered a high-risk plant. The previous method excludes the number of chemical facilities and simply evaluated the risk by focusing on the worst-case scenario [11].

The prior risk analysis method was an evaluation that considered risk severity, but not risk probability. In particular, under the previous OCA system and the recently implemented POCA system, large companies are still considered high-risk, since their facilities that include storage tanks, distillation columns, and others are large.

The key part of this study is small- and medium-sized businesses that were judged to be high-risk under the previous system. We assessed whether the K-risk assessment in the POCA system lessened the burden of excessive regulations and resulted in a reasonable safety system.

For this assessment, plating and paint manufacturing industries were selected. They are representative small- and medium-sized industries. Each business site formerly judged to be high-risk or medium-risk by the OCA criteria was examined. Their risk levels in the revised POCA were compared and analyzed to determine whether the new regulation is reasonable (Table 1).

### 2.3. Risk Analysis Methodology

#### 2.3.1. OCA Risk Analysis Methodology

In the OCA, scenarios were drawn for all facilities handling hazardous chemicals inside the plants. However, there is a standard for handling chemicals in small quantities for each chemical. Facilities that handle chemicals below this standard are excluded from the scenario [11].

A scenario was created assuming that a chemical accident occurred using a modeling program for each facility handling each chemical. The risk was calculated for each scenario as follows:
Risk = number of residents in the scenario × accident frequency(1)

The number of residents in the scenario is calculated by calculating the number of residents inside by drawing a circle centered on the source of the leak with the distance to the end point as the radius. In this case, the evaluation standard values of the endpoint [12] were
Toxic substance concentration: ERPG-2 (Emergency Response Planning Guidelines), AEGL-2 (Acute Exposure Guideline Levels), PAC-2 (Protective Action Criteria for Chemicals), etc.Radiant heat: 5 kW/m2 (40 s)Overpressure: 1 psi

Accident frequency is analyzed using the data of each of the following 1~3.
Reliability data are prepared by establishing data on accidents and breakdowns of equipment and facilities at the plant.Failure frequency data are provided by the equipment manufacturer.Reliability data of the following documents or data equivalent or higher include:
Offshore Reliability Data HandbookEuropean Industry Reliability Data BankNonelectronic Parts Reliability Data

A representative applicable theory of accident frequency uses the layer of protection analysis (LOPA) technique [13]. Representative figures of accident frequency by chemical facility in the LOPA are summarized in Table 2.

The risk is lowered by selection of an initiating event for each chemical facility’s characteristics and applying a facility that can be prepared at the factory when the event occurs. For example, in the case of a business that handles high-pressure vessels, a pressure vessel failure event may occur. If there is no safety device, the risk level is 10^−6^. However, in the case of a water curtain or relief valve, 10^−2^ is applied and the final risk is 10^−8^. The final risk is determined by considering the applicable safety devices for each facility and possible chemical accident events. Taking one pressure vessel facility as an example and applying the risk analysis, the results are as summarized in Table 3.

After estimating the risk of each facility, the final risk is presented by multiplying it by the number of residents within the scenario area. The risk of the plant itself is evaluated as high, medium, and low based on the worst scenario. The regulation of handling facility safety diagnosis is then applied according to the high, medium, and low grades [14]. On-site safety inspections are performed by the Ministry of Environment. High-risk plants are obliged to conduct a safety inspection once every four years, medium-risk plants once every 8 years, and low workplaces once every 12 years.

#### 2.3.2. Risk Analysis Methodology in POCA

The risk analysis methodology in the POCA proceeds in the same way as in the OCA for individual facilities. However, the risk assessment for the high-, medium-, and low-risk levels of the plant is presented in detail. This is different from the previous method of judging high-, medium-, and low-risk based on one worst-case scenario. This is an improvement from the previous problems of too many high-risk plants in the OCA, and the same high-risk classification applied to small and medium-sized businesses and major companies. The main improvements are as follows:
Consideration of the total number of scenarios: the level of risk is considered according to how many types of facilities there are.Consideration of the total sum of all LOPA risk levels for each scenario: even in a plant with a large number of facilities and scenarios, the risk can be lowered if the plant has safety facilities (dikes, relief valves, detector systems, etc.) to prevent chemical accidents.Application of the sum of the straight-line distances of the scenario where the radius is outside the site boundary of the business site is applied: the total number of scenarios that affect the environment outside the plants are analyzed.Calculation of the sum of the number of nearby residents within the scenario area by applying the score for each item. The scores are summarized in Table 4.

After calculating the score for each section, dividing the horizontal axis (accident frequency score) and vertical axis (accident impact score), and applying it to the risk assessment table, the final risk level grade (A/B/C) of the workplace can be confirmed. In this case, the accident frequency score is the sum of the number of accident scenarios and the accident scenario facility frequencies. The accident impact score is the sum of the accident scenario distances and the scores of the sum of the number of residents within the accident scenario. The workplace grade according to the risk assessment table is depicted in Figure 2.

However, the grade determined by the plant and final risk grade may differ. This is finally decided by the Chemical Safety Agency of the Ministry of Environment by an additional review of environmental factors around the plant. For example, for a plant with the same risk, if there is a residential complex nearby, or if an environmental protection target (such as a national park or water supply protection area) is included, the section score can be increased. In addition, the risk level may be lowered if a separate safety assurance plan that can reduce the risk is presented for the accident scenario facility that has the worst scenario on the risk at the plant [15].

When the risk is finally determined, the existing high-risk is grade A, medium-risk is grade B, and low-risk is grade C. Depending on the grade, a safety inspection must be performed once every four years for high-risk facilities, once every eight years for medium-risk facilities, and once every 12 years for low-risk facilities.

### 2.4. Accident Scenario Methodology

The scenario methodology used to estimate the number of residents within the chemical accident modeling range, which is one of the risk factors, is equally applied to the OCA and POCA. The Korea Off-site Risk Assessment (KORA: Ministry of Environment, Seoul, Korea) support tool developed by the Ministry of Environment is mainly used. In addition to this modeling program, the Areal Location of Hazardous Atmospheres (ALOHA: Environmental Protection Agency, Washington, DC, USA) program of the US EPA is applicable. There are several factors for deriving scenarios [16].

#### 2.4.1. Meteorological Element

The annual average meteorological factor in which the business site is located can be applied. Average temperature, average wind speed, average wind direction, and relative humidity act as meteorological factors during modeling. In addition, atmospheric stability, urban topography, and rural topography are included as additional factors [17]. The atmospheric stability of the Pasquill classes is summarized in Table 5.

#### 2.4.2. Operating Condition

The facilities subject to scenario modeling all handle hazardous chemical substances above the standard concentration in the plant. The difference between the OCA and POCA is the set of boundary lines for facilities that fall below the hazardous chemical standard concentration. In the OCA, even if a hazardous chemical is no longer a hazardous chemical due to a synthesis reaction after reaching the facility, the amount handled at the moment of input is calculated and the facility is targeted. In contrast, in the POCA, facilities that house chemicals that have reacted and are no longer considered to be hazardous chemicals were excluded from the scenario [15].

If the scenario was modeled on the assumption that 90% of the capacity of chemical facilities were handled in the OCA report, POCA was forcibly treated as 100% capacity of chemical facilities. This shows the intention of the regulation to evaluate the risk of plants for hazardous chemicals that are conservatively certain [18].

## 3. Results

### 3.1. Comparison of Worst-Case Scenarios for Each Plants

Risk levels determined before and after implementation of POCA, using the worst-case scenarios for each site were compared. The plating industry handles a large amount of hazardous chemicals, including hydrochloric acid, nitric acid, and sodium hydroxide, in the manufacture of plating solutions. In the paint manufacturing industry, solvents that include toluene, xylene, and methyl ethyl ketone, are used as raw materials to manufacture resins using pressure vessels, or paints are manufactured using mixers. A comparison of the worst-case scenarios for each business site is as follows. The chemical accident prevention system in Korea divides chemical accident cases into toxic leak accidents, fire and explosion accidents. There are five cases: toxic leak, pool fire, jet fire, vapor cloud explosion (VCE), and boiling liquid expanding vapor explosion (BLEVE).

In the case of companies A and B, both toxic and fire/explosion accidents can occur because of the nature of the industry and the materials handled. However, only toxic substance leakage accidents occur in the case of companies C and D, which mainly handle acids and bases (Table 6).

In the worst-case scenario, the most risky facility for all four study sites were the storage tanks. A storage tank is the beginning of the manufacturing process for any business. Maintenance is particularly important when using a power machine, such as a pump, and a safety device to prevent leakage is essential, especially with a dike. Company A’s scenario range is relatively small because it has an underground storage tank. For underground storage, the scenario range is relatively low in the event of a chemical accident. However, secondary problems, such as soil contamination, may occur after an accident (Figure 3).

### 3.2. Comparison of Chemical Facilities for Each Plant

Unlike the previous OCA, even for plants of similar industries and sizes, the number of facilities that can cause chemical accidents in POCA also affects the risk assessment of the plant. In other words, in the OCA system, the risk level of the plant is judged solely according to the worst-case described in Section 3.1. According to Table 1, in the OCA system, company A is judged as medium-risk, company B as high-risk, company C as medium-risk, and company D as high-risk. To analyze each plant in a more detailed manner in the revised POCA system, Table 7 compares facilities that are capable of causing chemical accidents, but which have a scenario range outside, but not inside, the plant.

### 3.3. Comparison of Risk Levels of Target Plants

The factors for determining the final risk rating of the plant included the accident frequency score and accident impact score. The accident frequency score is expressed as the sum of the number of accident scenarios (the larger the number of facilities indicating the scenario range) and the frequency of initiation event failure. The frequency of initiation event failures depends on the type of equipment used. For example, in the case of storage tanks, according to Table 3, the possible failure cases were selected and the score was calculated. The accident impact score is determined according to the interval score of the accident scenario distance (how large the scenario range is around the plant) and the interval score of the number of residents within the scenario area (the number of people living in the area). The detailed score table for each accident frequency score and impact score is presented in Figure 4 [19].

The final risk grade of the four plants in this study was determined to be “C”. Companies A and C were in the middle grade, and companies B and D had much lower grades compared to the previous system, which assigned high-risk grades. This is a much more realistic classification of plants than simply selecting one worst-case accident scenario. These four plants are mid-sized companies in Korea. Company B and Company D were managed as large high-risk companies with the same level of risk as the major companies. This also raises a number of issues related to equity. In addition, simply selecting the level of risk based on the worst-case scenario without considering the number of facilities can be seen as excessive regulation of plants. The risk classification table for each business site is depicted in Figure 5.

## 4. Discussion

This study examined the history of changes in the chemical accident prevention system in Korea, and conducted a risk analysis of plants affected by the system implemented in 2021. In the previous system, plants that handled even a very small amount of hazardous chemicals, such as laboratories and hospitals, were required to submit an OCA report. The prevailing opinion was that this was excessive regulation [20].

In the early stages of the implementation of the Chemicals Control Act, it was necessary to implement regulations, including all plants, to prevent chemical accidents and ensure public safety. However, five years after the establishment of such a system, it was necessary to consider and change the regulations. Accordingly, in 2021, the Ministry of Environment introduced a chemical accident prevention system to eliminate plant regulations. Legislation was introduced to determine and manage plant risks more rationally.

The level of risk in plants is important because of the safety examination conducted by the state. Safety inspection is an evaluation conducted by the Korea Ministry of Environment for all plants. It is obliged to do this once every 4 years for grade A, once every 8 years for grade B, and once every 12 years for grade C [21].

In the existing OCA system, companies A and C were medium-risk and companies B and D were high-risk. Companies A and B handle similar chemicals (solvents that include toluene and methyl ethyl ketone) in the same paint manufacturing industry. The scale of reactor and mixer holdings is similar, except for underground and above-ground storage tanks. However, in the revised system, all were judged to be grade C (low-risk). The risk assessment of the two plants showed that the accident impact score was higher than the accident frequency score.

This means that the impact of the accident is large, with many residents in the immediate vicinity. In the case of these plants, cooperation with local residents and local governments is more important in the event of a chemical accident than preventing the occurrence of a chemical accident [22]. The OCA did not contain any information on cooperation between local residents and local governments.

In particular, if there are many local residents, it is necessary to quickly communicate news of a chemical accident to these residents and implement countermeasures. Additionally, in the case of plant A, the scenario caused by toxic leakage from the underground storage tank is greater than the damage caused by fire and explosion. Therefore, it is possible to prevent toxic leakage by providing a large amount of protective masks to residents in the plants. Conversely, in the case of plant B, because the fire and explosion scenario is the worst, countermeasures need to address chemical accidents through periodic inspection of fixed fire extinguishing equipment and/or establishing and training a fire brigade.

Sites C and D also had higher accident impact scores than the accident frequency scores. However, owing to the characteristics of the materials handled (i.e., many acids and bases), accidents due to toxic leakage are a danger. Therefore, it is important to have gas masks or absorbent cloth available for initial countermeasures. In particular, in the case of plant C, the accident frequency score was one point higher than that of plant D. This suggests that there is a need for a safety facility for equipment related to the frequency of initiation event failures. In other words, even if the plants C and D have the same storage tank or plating tank, plant D should install and manage a dike to prevent leakage, have a fire extinguishing facility, and have a leak detector. Plant C is insufficient compared to plant D. In the case of plant C, chemical accidents can be prevented by installing safety facilities (leak prevention facilities, leak detection facilities, etc.) to increase the safety level [23].

These systems are limited because chemical accident cases are not considered. Comparing toxic leaks and fire explosions on the same line is problematic. Fire and explosion accidents are more dangerous than toxic leaks, and many people can be injured [24]. In particular, fire damage can spread to nearby businesses if repeated explosions occur, rather than just one explosion, especially in the case of a plant with a risk of fire explosion compared to plants without this risk. Thus, additional points should be given to the level of risk [25].

In the follow-up study, it is necessary to review whether the system is applied to the petroleum refining industry or the chemical product manufacturing industry, which handles chemicals with a risk of fire and domino explosion and produces products through high-temperature and high-pressure processes.

## 5. Conclusions

This study examined the history of changes in the chemical accident prevention system in Korea and its applicability to low- and medium-sized businesses. Chemical accidents can occur in any chemical industry. In addition, chemical accidents may not just material and human losses at the workplace, but can extend to the surrounding environment and local residents.

To prevent such accidents, advanced countries, including Korea, have established a national preventive system to manage their plants. In Korea, the Chemicals Control Act was enacted in 2015 in response to the hydrofluoric acid accident in Gumi in 2012. The legislation was important. It mandated the inspection of chemical facilities owned by each plant and the regulated management of the most dangerous chemical facilities through chemical accident modeling.

However, following the implementation of the Act, the need for equity at each business site became apparent, such as at plants with many chemical facilities and only one chemical facility with large modeling results. Accordingly, the 2021 chemical accident prevention plan system introduced various factors into the risk analysis for each business site. A Korean-style risk analysis was also introduced to present reasonable regulations.

In the present study, four plants were selected for research, and their risks were analyzed. The results revealed that all four plants were subjected to excessive regulation under the previous system. Compared to the previous system, the revised chemical accident prevention plan is a comprehensive plan concerning deficiencies in the plants and areas that need to be improved. Chemical accidents can occur at any time, and it is important to manage plants by comprehensively evaluating the risks. This Korean-style risk analysis could be applied in other countries that have chemical industries. The regulations and indicators could prevent chemical accidents.

In addition, in order to prevent chemical accidents and achieve sustainable development, it is necessary to conduct regulatory research on the Korean-style customized chemical industry [26].

## Figures and Tables

**Figure 1 ijerph-18-11982-f001:**
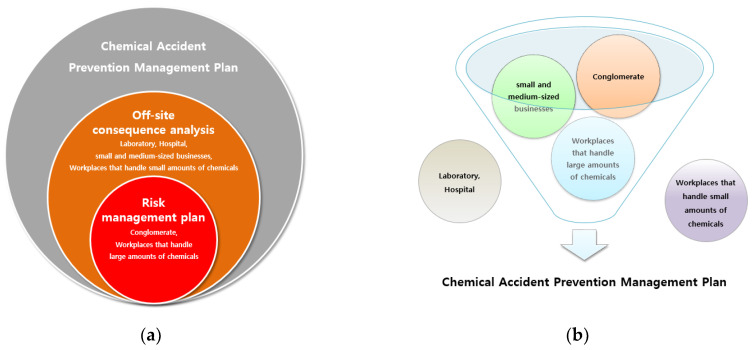
Changes in the chemical accident prevention system in Korea. (**a**) Plants subject to OCA and RMP from 2015 to 2020; (**b**) Plants in POCA legislation implemented in 2021.

**Figure 2 ijerph-18-11982-f002:**
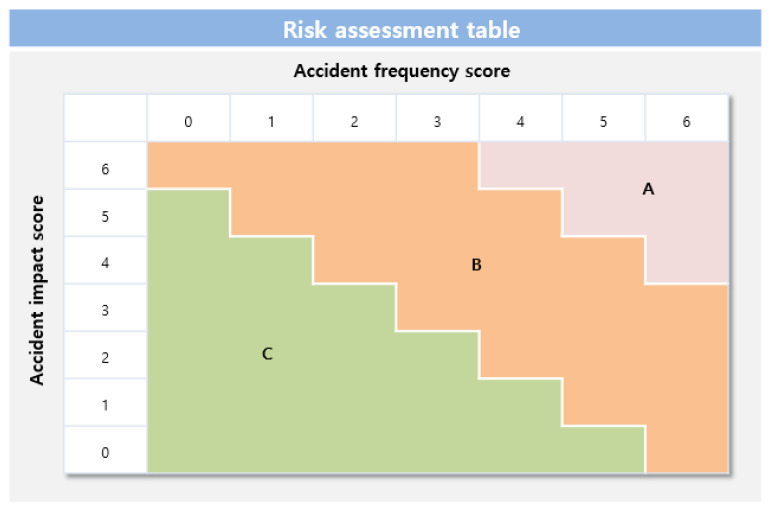
Risk rating determination scorecard.

**Figure 3 ijerph-18-11982-f003:**
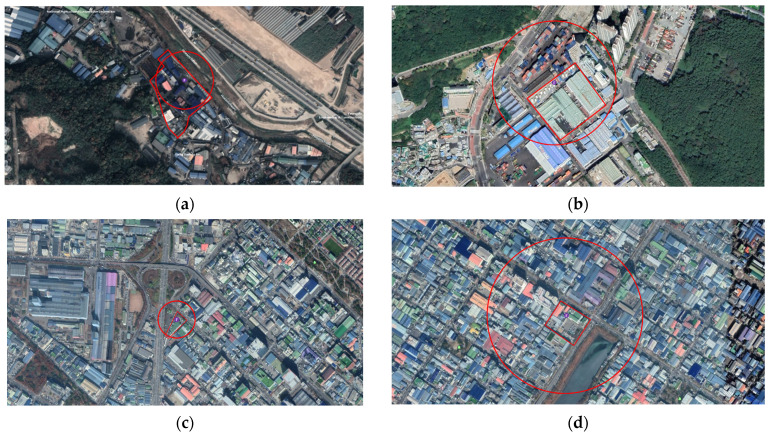
Worst-case scenario range of plants. (**a**) Company A: toluene storage tank toxic leak. (**b**) Company B: methyl ethyl ketone storage tank VCE. (**c**) Company C: nitric acid storage tank toxic leak. (**d**) Company D: ammonia water storage tank toxic leak.

**Figure 4 ijerph-18-11982-f004:**
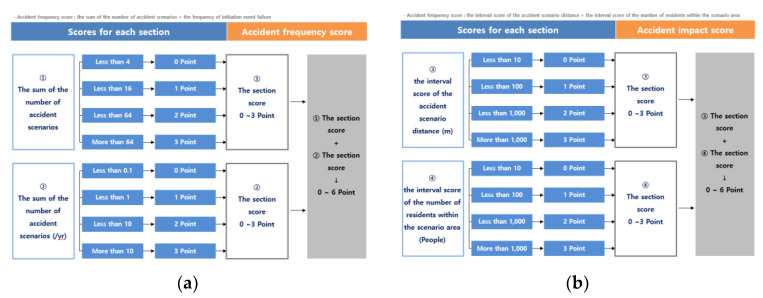
Plant risk rating matrix scores. (**a**) Accident frequency score and (**b**) Accident impact score.

**Figure 5 ijerph-18-11982-f005:**
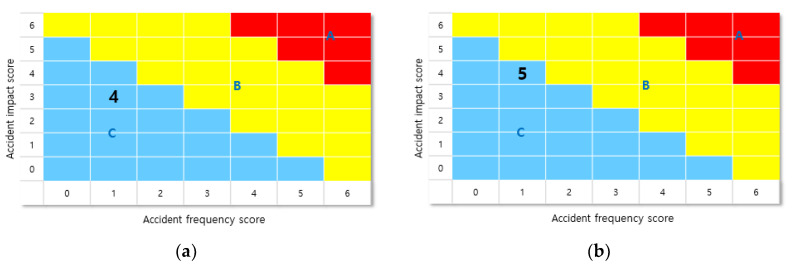
Final risk levels. Grade of (**a**) Company A; (**b**) Company B; (**c**) Company C, and (**d**) Company D.

**Table 1 ijerph-18-11982-t001:** Characteristics of the target plants and types of chemicals handled.

Plant	Industry Type	Size	Hazardous Chemicals Used	Special Features
A	paint manufacturing	mid-sized (54 employees; annual sales $0.8 billion)	Toluene, Xylene	Mainly manufactures paint for building interiors. Chemicals held in underground storage tanks. OCA medium-risk plant
B	paint manufacturing	mid-sized (150 employees; annual sales $1.2 billion)	Toluene, Methyl ethyl ketone	Mainly manufactures anti-rust paint for ships and has more than 10 solvent tanks. OCA ^1)^ high-risk plant
C	plating	mid-sized (95 employees; annual sales $3.5 billion)	Hydrochloric acid, Nitric acid	Directly manufactures plating solution, or plates the contact material using silver with the manufactured plating solution. OCA ^1)^ medium-risk plant
D	Plating	mid-sized (79 employees; annual sales $2.3 billion)	Ammonia water, Sodium hydroxide	Industrial high-purity precious metal plating by manufactured aqua regia. OCA ^1)^ high-risk plant

OCA: off-site consequence analysis.

**Table 2 ijerph-18-11982-t002:** LOPA initiating event value per year.

	Initiating Event	Example of a Value Chosen by a Company for Use in LOPA (per Year)
1	Pressure vessel failure	10^−^^6^
2	Piping rupture failure per 100 m	10^−^^5^
3	Piping leak per 100 m	10^−^^3^
4	Atmosphere tank failure	10^−^^3^
5	Flange/valve leak	10^−^^3^
6	Pump/compressor leak	10^−^^3^
7	Premature opening of spring-loaded relief valve	10^−^^2^
8	Cooling water failure	10^−^^1^
9	Unloading/loading hose failure	10^−^^2^
10	External fire	10^−^^2^

LOPA: Layer Of Protection Analysis.

**Table 3 ijerph-18-11982-t003:** LOPA risk assessment of pressure vessels.

	Initiating Event	Example of a Value Chosen by a Company for Use in LOPA (per Year)	Count	Probability of Failure on Demand		Level of Risk
1	Pressure vessel failure	10^−^^6^	1	Water curtain (10^−^^1^)	Relief valve (10^−^^2^)	10^−^^9^
2	Piping rupture failure per 100 m	10^−^^5^	1	Detector and shut-off valve (10^−^^2^)		10^−^^7^
3	Piping leak per/100 m	10^−^^3^	1	Double piping (10^−^^1^)		10^−^^4^
5	Flange/valve leak	10^−^^3^	20	Dike (10^−^^2^)		20 × 10^−^^5^
6	Pump/compressor leak	10^−^^3^	1	Seamless pump (10^−^^1^)		10^−^^4^
7	Premature opening of spring-loaded relief valve	10^−^^2^	1	Relief vale/Rupture disk (10^−^^2^)		10^−^^4^
8	Cooling water failure	10^−^^1^	1	Stand by pump (10^−^^1^)		10^−^^2^
10	External fire	10^−^^2^	1	Fire proofing (10^−^^1^)	Sprinkler (10^−^^1^)	10^−^^3^
Total	Σ(LOPA valve)×Probability of failure on demand					1.15 × 10^−^^2^

LOPA: Layer Of Protection Analysis.

**Table 4 ijerph-18-11982-t004:** Section score for risk determination.

Section Score (Points)	Sum of the Number of Accident Scenarios	Sum of Accident Scenario Facility Frequency per Year	Sum of Accident Scenario Distances (m)	Sum of the Number of Residents within the Accident Scenario
0	less than 4	less than 0.1	less than 10	less than 10
1	less than 16	less than 1	less than 100	less than 100
2	less than 64	less than 10	less than 100	less than 100
3	64 or more	10 or more	1000 or more	1000 or more

**Table 5 ijerph-18-11982-t005:** Meteorological stability for Pasquill classes.

Wind Speed (m/s)	Day			Night	
	Radiation Intensity				
	Strong	Cloudy	Sunny	Cloudy	Sunny
<2	A	A–B	B	F	F
2–3	A–B	B	C	E	F
3–5	B	B–C	C	D	E
5–6	C	C–D	D	D	D
>6	C	D	D	D	D

A: Very unstable; B: instability; C: slight instability; D: neutral; E: slightly stable; F: very stable.

**Table 6 ijerph-18-11982-t006:** Worst-case scenario by plants.

Plant	Industry	Worst-Case Scenario Target Facility	Worst-Case Scenario Type	Scenario Range (Radius in m)
A	Paint manufacturing industry	Toluene storage tank	Toxic accident case	57.5
B	Paint manufacturing industry	Methyl ethyl ketone storage tank	Vapor cloud explosion	146.7
C	Plating industry	Nitric acid storage tank	Toxic accident case	159.6
D	Plating industry	Ammonia water storage tank	Toxic accident case	402.3

**Table 7 ijerph-18-11982-t007:** Number of facilities capable of causing chemical accidents scenario for risk analysis.

Plant	Number of Storage Tanks	Number of Plating Tanks	Number of Reactors	Number of Mixers
A	4	Not applicable	3	6
B	2	Not applicable	3	6
C	2	6	Not applicable	2
D	2	8	Not applicable	Not applicable

## Data Availability

Not applicable.

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
