# Peer review of "Changes in Risk in Medium Business Plating and Paint Manufacturing Plants following the Revision of the Korean Chemical Accident Prevention System"

_ijerph, 2021, doi:10.3390/ijerph182211982_

Round 1
Reviewer 1 Report
Dear authors,
many thanks for your article. It was a pleasure for me to review it.
I consider the article to be up to date, but I would suggest to revise and shorten the title a bit to become more attractive for the readers.
with best regards
Author Response
Point 1
I consider the article to be up to date, but I would suggest to revise and shorten the title a bit to become more attractive for the readers.
Response 1
We changed the title to be shorter and clearer. ‘Changes in risk in medium business plating and paint manufacturing plants following the revision of the Korean Chemical Accident Prevention System’
Thank you for your review. These reviews have contributed to the quality of this paper.

Reviewer 2 Report
Dear Authors, first of all let me express my compliments for the interesting subject of your research. I do not have serious negative considerations about your paper in its structure, results showed and analysis about them.
I think the paper is interesting and it could be considered for publication. However, this paper can be further improved in the following points.
- the English level should be improved.
- the "Introduction" should be shorten or may be divided in two section (introduction and literature review);
- "future works" should be better explained
Please improve english and references, e.g.
- Green chemistry contribution towards more equitable global sustainability and greater circular economy: A systematic literature review Silvestri, C., Silvestri, L., Forcina, A., Di Bona, G., Falcone, D. this
Author Response
Point 1
The English level should be improved.
Response 1
The entire English text has been corrected.
Point 2
The "Introduction" should be shorten or may be divided in two section (introduction and literature review);
Response 2
The introduction part is divided into two sections. (Line 41, 61) This shows the history of chemical accidents and changes in the chemical accident prevention systems in the US and Korea. These contents suggest the need for appropriate revision of the chemical accident prevention system, and how the changed system in Korea was customized and applied to Korean companies.
Point 3
"future works" should be better explained
Response 3
The necessity and implications of follow-up studies to be performed in the future have been added to the discussion section.(Line 453~456)
Point 4
Please improve english and references, e.g.
Green chemistry contribution towards more equitable global sustainability and greater circular economy: A systematic literature review Silvestri, C., Silvestri, L., Forcina, A., Di Bona, G., Falcone, D. Journal of Cleaner Productionthis 2021, 294, 126137
Response 4
It is added to the conclusion that the government's continuous research and researchers' efforts are necessary for sustainable green chemicals. It suggested that continuous benchmarking of research in advanced countries is necessary for a customized system in Korea. (Line 482~484)
Thank you for your review. These reviews have contributed to the quality of this paper.

Reviewer 3 Report
The article “Changes in Risk in Medium Business Plants Following Revision of the Korean Chemical Accident Prevention System, Focusing on Plating and Paint Manufacturing Industries” compares the overall risk of plants in the chemical accident prevention system, the 2015 version of the off-site consequence analysis, and the newly implemented Preparation of Off-Site Consequence Analyses system. The feasibility of the new system for small- and medium-sized plants was evaluated, and limitations were identified. The prior risk analysis method was an evaluation that considered risk severity but not risk probability.
Presented material seems to be rather a literature review than a research report. The presented study for four plants is not detailed enough and clear. There are no clear conclusions from this study presented too.
Moreover, the article is not necessarily so long. There is much information repeated, especially in the discussion and conclusions chapters.
The detailed remarks are as follows:
- The abstract should include information on what exactly the article presents,
- There is not described in the text what Fig. 1 presents.
Generally, the article should be rebuilt to make it more clear and give any general recommendations.
Author Response
Point 1
Presented material seems to be rather a literature review than a research report. The presented study for four plants is not detailed enough and clear. There are no clear conclusions from this study presented too.
Moreover, the article is not necessarily so long. There is much information repeated, especially in the discussion and conclusions chapters.
The detailed remarks are as follows:
1.The abstract should include information on what exactly the article presents,
Response 1
Korea has used various regulations related to chemical substances by benchmarking the systems of the US and EU, which are mostly developed countries. In particular, most of the chemical accidents that occurred in Korea were related to occupational diseases of workers, rather than the type that affected the surrounding environment and nearby residents like US and EU. However, due to the hydrogen fluoride accident in Gumi in 2012, it was revealed that accidents related to chemicals may affect local residents. Accordingly, in 2015, Korea benchmarked the US RMP system and created the OCA system. However, since this system is tailored to large American factories, it was considered excessive regulation by many small and medium-sized plants in Korea. Accordingly, in 2021, Korea developed and applied its own risk assessment. This has implications for the development of a Korean-style risk assessment tool that goes one step further from the existing systems of advanced countries. Accordingly, the value of this study is important, and furthermore, it has significance as an introduction and feasibility assessment of a risk assessment system applicable to developing countries with many small and medium-sized plants worldwide.
In the abstract, the contents of the risk change, which is the most important part of this study, have been revised in detail. (Line28~33)
Point 2
2.There is not described in the text what Fig. 1 presents.
Response 1
An explanation for Figure 1 has been added. (Line 153~157)
Thank you for your review. These reviews have contributed to the quality of this paper.

Reviewer 4 Report
A clear and very well written paper on an important societal topic. No further comments.
Author Response
Thank you for your review. These reviews have contributed to the quality of this paper.
Round 2
Reviewer 3 Report
In my opinion, the suggested corrections were not implemented enough.
Presented material continuously seems to be rather a literature review than a research report. There are still no clear conclusions from this study, which could be used as a general recomendations for similar cases to the cases described in the examples.
Moreover, the article is not necessarily so long. There is much information repeated, especially in the discussion and conclusions chapters.
The revision doesn't clearly present which changes have been implemented. It should be done more clearly, which sentences/words are new and which are deleted.
Author Response
Point 1
Presented material continuously seems to be rather a literature review than a research report. There are still no clear conclusions from this study, which could be used as a general recomendations for similar cases to the cases described in the examples.
Response 1
This study is about the chemical accident prevention system and risk assessment in Korea. Korea has made efforts to prevent chemical accidents by following the systems of the United States and Europe. However, the systems of developed countries have limitations that do not match the reality of Korea. Korea has a narrow land and has undergone difficult development due to rapid economic growth. The separation of factories and residential areas has not been clearly achieved, and such problems persist to this day. On the other hand, in the US and Europe, factories and residential areas are clearly separated and operated. In this regard, there is a limit to introducing the chemical accident prevention system of the US and Europe as it is in Korea.
In 2021, a Korean-style risk assessment was introduced to re-evaluate the risks of chemical plants. As a result, all factories that had received A and B grades through the existing American and European methods were rationally changed to C grade. Through such continuous research, it is expected that the Korean risk assessment can be applied to Hong Kong and China in Asian regions with similar development characteristics. Reviewers should consider these points when evaluating the article.
Point 2
Moreover, the article is not necessarily so long. There is much information repeated, especially in the discussion and conclusions chapters.
Response 2
Duplicate descriptions in the introduction have been deleted from the discussion. (Line 389~399)
Thank you for your review. These reviews have contributed to the quality of this paper.
